# HC-BDC: Human Cognition-Inspired Bayesian Distribution Calibration for Few-Shot Classification

## Abstract

The fundamental challenge of few-shot image classification stems from inadequate distributional representations due to limited training samples. This paper presents a **H**uman **C**ognitive-Inspired **B**ayesian **D**istribution **C**alibration method (HC-BDC), inspired by human **fast and slow thinking** and **the neurocognitive mechanisms of convergent and divergent thinking**. Unlike conventional approaches, our framework implements a dual-phase reasoning mechanism: the fast-thinking phase employs a lightweight Mixture-of-Experts (MoE) model to dynamically allocate existing knowledge for different few-shot tasks, while the slow-reasoning phase utilizes Bayesian relation inference to simulate human convergent and divergent thinking. This approach diversely generates associations between novel concepts and prior knowledge from multiple perspectives, leading to more comprehensive distribution representations. Specifically, the fast-thinking process automatically selects relevant knowledge components through attention routing, whereas the slow-reasoning process constructs multi-view relational graphs via Bayesian inference to dynamically capture diverse inter-class relationships. Extensive experiments on multiple benchmark datasets demonstrate that our approach outperforms current state-of-the-art methods. The HC-BDC framework provides a novel direction for interpretable few-shot learning by modeling the interaction between unconscious association and conscious reasoning processes.

## 1 Introduction

While deep learning has achieved remarkable success in computer vision through leveraging large-scale labeled data Yan et al. (2016); Lin et al. (2020), its performance in data-scarce scenarios for few-shot image classification remains challenging. Current few-shot learning approaches includes meta-learning, metric learning, and data augmentation Wang et al. (2020). Most methods examine class relationships from a single perspective and rely on fixed similarity measures (e.g., Euclidean distance Yang et al. (2021) or parameter-frozen linear layers Vinyals et al. (2016)) for relationship computation. These limitations constrain the performance of these models on few-shot tasks. In contrast, humans demonstrate exceptional capability in learning novel concepts from limited examples by employing dual thinking modes of fast and slow thinking Kahneman (2011), along with the ability to divergent and convergent thinking Zhang et al. (2020).

Inspired by these findings, we deconstruct human thinking mechanisms into two systems: fast thinking system and slow reasoning system, each incorporating elements of divergent exploration and convergent integration. Our framework introduces two key innovations: (1) a lightweight gated MoE that dynamically routes base-class knowledge via attention mechanisms, simulating fast-thinking selection; and (2) a Bayesian relation inference module constructing multi-view Gaussian graphs through probabilistic sampling, emulating slow-reasoning exploration and integration. Specifically, we present the **H**uman **C**ognitive-Inspired **B**ayesian **D**istribution **C**alibration (HC-BDC) framework, a Bayesian relation inference approach with three main contributions:

- **MoE-based Dynamic Knowledge Routing**: We design a lightweight MoE component that dynamically and divergently routes input samples to specialized experts through atten-

tion mechanisms, simulating how humans flexibly associate novel instances with diverse existing knowledge categories in fast thinking system.

- **Bayesian Multi-View Relation Modeling**: We develop a Bayesian relation inference module that constructs multi-view Gaussian relational graphs through probabilistic sampling, emulating human divergent and convergent thinking to adaptively capture and consolidate inter-class relationships in slow reasoning system.

- **Interpretable Cognitive Alignment**: The framework demonstrates meaningful consistency between the generated relational graphs and human reasoning patterns through visual analysis, offering essential explainability for applications such as medical diagnosis.

Extensive experiments demonstrate our method's superiority over state-of-the-art approaches across multiple benchmarks. The proposed cognitive simulation mechanisms provide a new direction for developing human-like learning systems, particularly valuable for scenarios requiring both data efficiency and interpretability.

## 2 RELATED WORKS

Few-shot learning aims to acquire correct knowledge from very limited samples. Existing methods can be broadly categorized into model-based, metric-based, and data augmentation approaches Wang et al. (2020).

Model-based methods use transfer learning; fine-tuning techniques Nakamura & Harada (2019) adapt pre-trained networks but face catastrophic forgetting. Meta-learning solutions Finn et al. (2017); Rusu et al. (2018) simulate few-shot tasks during training, yet often incur high computational costs.

Metric-based paradigms learn discriminative embeddings. Prototypical Networks Snell et al. (2017) use class prototypes, while Matching Networks Vinyals et al. (2016) apply attention-based similarity. However, these rely on static prototype matching and fail to capture multi-view inter-class relations. Recent relation-aware methods Chen et al. (2019); Yang et al. (2021) model class dependencies but still lack dynamic, context-aware reasoning as in human cognition Zhang et al. (2020).

Data augmentation techniques address sample scarcity through feature space enrichment, where semi-supervised methods Ren et al. (2018a) leverage unlabeled data and contrastive learning Yang et al. (2022) enhances discriminability. While effective for instance-level generalization, these approaches lack mechanisms for modeling the hierarchical knowledge organization and multi-perspective reasoning characteristic of human cognition Zhang et al. (2020). More recently, Yang et al. (2021) introduce distribution calibration using Euclidean distances between class statistics, while Wei et al. (2023) develop direction-driven weighting for feature distribution fitting. However, compared with human thinking patterns, these works still exhibit certain deficiencies: (1) their relational thinking modes are singular, whereas humans possess complementary fast and slow thinking systems that enable more comprehensive and efficient relational inference Kahneman (2011); (2) their relation generation processes remain single-viewed, failing to capture the diverse potential associations humans naturally consider during few-shot learning.

In cognitive science, human thinking is divided into two systems: fast (rapid, intuitive) and slow (deliberate, analytical) Kahneman (2011). Furthermore, human creative cognition is increasingly understood through the dual mechanisms of divergent and convergent thinking Zhang et al. (2020). These processes are regulated by metacontrol states that bias cognition toward flexibility (broad associative thinking) or persistence (focused problem-solving) Hommel (2015); Mekern et al. (2019). Such neural dynamics facilitate the multi-perspective relational reasoning and dynamic knowledge integration that underlie human-like learning and creativity Kounios & Beeman (2014). These insights provide a foundational framework for developing models that more accurately emulate human cognitive processes.

Our proposed HC-BDC framework breaks through existing limitations by: (1) achieving dynamic knowledge retrieval through gated MoE mechanisms that simulate the selective attention process in human fast thinking, and (2) replacing static metrics with probabilistic multi-view relation graphs that emulate the divergent-convergent thinking in human slow reasoning processes, thereby enabling

multi-perspective relational reasoning. This dual innovation bridges the gap between existing AI models and cognitive science principles in few-shot learning.

# 3 METHOD

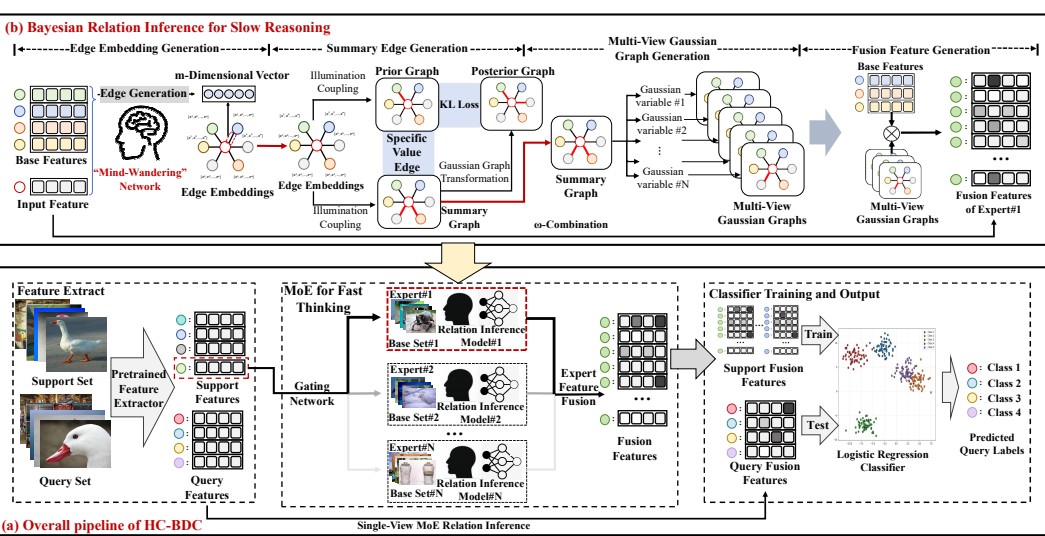

Figure 1: The overall architecture of the proposed Human Cognitive-Inspired Bayesian Distribution Calibration (HC-BDC) framework for few-shot learning.

## 3.1 PIPELINE OF HC-BDC

The overall pipeline of HC-BDC is shown in the lower part of Figure 1. The input of HC-BDC is an $n$-way $k$-shot few-shot learning episode. The episode includes labeled support set $\mathcal{S} = \{(\mathbf{x}_i, \mathbf{y}_i)\}_{i=1}^{n \times k}$ and unlabeled query set $\mathcal{Q} = \{\mathbf{x}_j\}_{j=1}^{m_q}$. First, we adopt a pre-trained feature extractor $f_\theta(\cdot)$ to extract features from the support and query images:

$$\mathbf{h}_i = f_\theta(\mathbf{x}_i), \quad \forall \mathbf{x}_i \in \mathcal{S} \cup \mathcal{Q}, \tag{1}$$

where $\mathbf{x}_i \in \mathbb{R}^{3 \times H \times W}$ represents an input RGB image with height $H$ and width $W$, and $\mathbf{h}_i \in \mathbb{R}^d$ is the extracted feature vector. Each extracted feature vector $\mathbf{h}_i$ is then processed by MoE fast thinking component (The Bayesian relation inference module for slow reasoning is invoked within this component.) to generate multiple fusion features that represent the underlying distribution:

$$\mathcal{F}_i = \text{MoE}(\mathbf{h}_i), \tag{2}$$

where $\mathcal{F}_i = \{\mathbf{f}_i^k\}_{k=1}^{N_{\text{Gaus}}}$ denotes the set of $N_{\text{Gaus}}$ fusion features generated for input $\mathbf{h}_i$. Details of the MoE component are described in 3.2. Each fusion feature $\mathbf{f}_i^k \in \mathbb{R}^d$ captures a distinct perspective of the input feature's distribution through multi-view relational reasoning. All fusion features generated by support features and their corresponding labels are used to train a logistic regression classifier:

$$\text{Classifier} \leftarrow \text{Train}_{\text{LR}}(\mathcal{F}_{\text{supp}}, \mathbf{y}_{\text{supp}}), \tag{3}$$

where $\mathcal{F}_{\text{supp}}$ represents all fusion features generated by support features and $\mathbf{y}_{\text{supp}}$ represents their labels. For each query set feature $\mathbf{h}_j$, we represent it using a single fusion feature $\mathbf{f}_j$ from one perspective and employ the trained classifier to perform the classification task.

$$\hat{y}_j = \text{Classifier}(\mathbf{f}_j). \tag{4}$$

## 3.2 MOE FOR FAST THINKING

The HC-BDC framework emulates human cognitive mechanisms Kahneman (2011) for few-shot distribution calibration. Our approach utilizes a Mixture-of-Experts (MoE) architecture to simulate fast thinking processes, integrating Bayesian relation inference for slow reasoning to generate

multi-view fusion features $\mathcal{F}$ (Eq. 2). The MoE component initiates with a gating network that implements divergent thinking through dynamic attention allocation across experts. Each expert maintains distinct base-class prototypes, which are selected from the training data and kept static during optimization. For each episode, support set features are aggregated to compute expert attention weights:

$$\boldsymbol{\alpha} = \text{GatingNetwork}(\mathcal{H}_{\mathcal{S}}), \tag{5}$$

where $\mathcal{H}_{\mathcal{S}} = \{\mathbf{h}_i\}_{i=1}^{n \times k}$ represents all support set image features, and $\boldsymbol{\alpha} = [\alpha_1, \alpha_2, \ldots, \alpha_{N_E}]^\top \in \mathbb{R}^{N_E}$ denotes the attention weight vector satisfying $\sum_{e=1}^{N_E} \alpha_e = 1$, with $N_E$ being the number of experts. The gating network employs the standard self-attention method from the Transformer encoder architecture for attention computation Vaswani et al. (2017).

For any input feature $\mathbf{h}_i$ in this episode, the model utilizes these attention weights for expert allocation. Within each expert, we invoke the Bayesian relation inference module in 3.3 to simulate slow reasoning system and generate multi-perspective fusion features. The final output combines all expert features through attention-weighted fusion:

$$\mathcal{F}_i^e = \text{BayesianModule}(\mathbf{h}_i, \mathcal{H}_b^e), \tag{6}$$

$$\mathcal{F}_i = \sum_{e=1}^{N_E} \alpha_e \cdot \mathcal{F}_i^e, \tag{7}$$

where $\mathcal{H}_b^e = \{\mathbf{h}_{b_j}^e\}_{j=1}^{N_B}$ represents the base-class prototypes for expert $e$, and $\mathcal{F}_i^e = \{\mathbf{f}_i^{e,k}\}_{k=1}^{N_{\text{Gaus}}}$ denotes the multi-perspective fusion features generated by expert $e$, $\mathcal{F}_i = \{\mathbf{f}_i^k\}_{k=1}^{N_{\text{Gaus}}}$ is the final set of multi-view fusion features output by the MoE fast thinking component.

### 3.3 BAYESIAN RELATION INFERENCE FOR SLOW REASONING

Human slow-reasoning systems are crucial for reducing judgment errors and improving decision reliability Kahneman (2011), while divergent and convergent thinking underpin creative cognition Zhang et al. (2020). Inspired by these mechanisms, we develop a Bayesian relation inference module to simulate slow reasoning for feature enhancement.

As shown in the upper part of Figure 1, each module incorporates $N_B$ base-class prototypes $\mathcal{H}_b$ from the training set, mimicking humans' structured prior knowledge. For an input feature $\mathbf{h}_i$, the module first establishes divergent associations with base classes, generating relational edge embeddings:

$$\mathbf{e}_{ij} = \phi([\mathbf{h}_i | \mathbf{h}_{b_j}^e]), \quad \phi : \mathbb{R}^{2d} \to \mathbb{R}^{d_e}, \tag{8}$$

where $|$ denotes concatenation and $\phi$ is a linear neural encoder to simulate human mind-wandering. A key to creative behavior is gathering ideas together through convergent thinking (illumination) Wallas (1926); Zhang et al. (2020). Therefore, we simulate human cognitive integration by coupling these edge embeddings into a single summary edge, representing the aggregated inter-class relational strength after synthesizing divergent associations. Recent research suggests that humans engage in numerous unconscious perceptions during relational thinking, which can be modeled as sampling from a binomial distribution with parameters $n \to \infty$ and $\lambda \to 0$ Huang et al. (2020). And the coupling process can be viewed as sampling from this distribution. Based on the De Moivre–Laplace theorem and prior studies Huang et al. (2020); Liu & Jia (2023), we transform this sampling into a Gaussian approximation via the following conversion (see the Appendix for justification):

$$\mu_{\text{ij}} = \zeta \left( \text{L}_{\text{mean}} \left( \mathbf{e}_{ij} \right) \right) + \epsilon, \tag{9}$$

$$\sigma_{\text{ij}} = \zeta \left( \text{L}_{\text{std}} \left( \mathbf{e}_{ij} \right) \right), \tag{10}$$

$$m_{\text{ij}} = \frac{1 + 2\mu_{\text{ij}}\sigma_{\text{ij}}^2 - \sqrt{1 + 4\mu_{\text{ij}}^2\sigma_{\text{ij}}^4}}{2}, \tag{11}$$

where $\zeta$ denotes the softplus activation function, $\epsilon$ is a small constant, and $\text{L}_{\text{mean}}(\cdot)$ and $\text{L}_{\text{std}}(\cdot)$ are neural networks that estimate the mean and standard deviation of the Gaussian distribution $\mathcal{N}(\mu_{ij}, \sigma_{ij}^2)$ approximating the binomial distribution $\mathcal{B}(n, \lambda)$ under $n \to \infty$, $\lambda \to 0$. The graph $\mathbf{M} = [m_{ij}]$ couples the cognitive relations between novel and each base class to a specific numerical value. We employ variational inference to train this module, thereby ensuring the stability of

the generated relational graph $M$. Specifically, we generate two coupled graphs using an identical architecture: one serves as the prior graph, while the other is referred to as the "summary graph". The summary graph is utilized for subsequent multi-view Gaussian graph generation.

To enhance the creative capacity of the component, we again emulate human divergent thinking Zhang et al. (2020) by transforming the summary graph into multiple Gaussian relational graphs via a multi-view Gaussian graph transformation. This process introduces Gaussian random variables associated with the edge weights of the summary graph, updating them from diverse perspectives. Each Gaussian graph is generated as follows for view $k$:

$$\widetilde{\alpha}_{\text{ij}} = \sqrt{m_{\text{ij}}(1.0 - m_{\text{ij}})} \cdot \widetilde{\varepsilon} + m_{\text{ij}}, \tag{12}$$

$$s_{\text{ij}}^k = m_{\text{ij}}^{\text{mean}} \cdot \widetilde{\alpha}_{\text{ij}} + m_{\text{ij}}^{\text{std}} \cdot \sqrt{\widetilde{\alpha}_{\text{ij}}} \cdot \varepsilon_{\text{ij}}^k, \tag{13}$$

$$\bar{\alpha}_{\text{ij}}^k = s_{\text{ij}}^k \cdot \widetilde{\alpha}_{\text{ij}}, \tag{14}$$

$$\alpha_{\text{ij}}^k = \zeta\left(\bar{\alpha}_{\text{ij}}^k\right), \tag{15}$$

where $\widetilde{\varepsilon}$ and $\varepsilon_{\text{ij}}^k$ are standard Gaussian variables, $\mathbf{S}^k = [s_{\text{ij}}^k]$ represents the edge-related Gaussian variable, $\zeta$ denotes the operation of normalizing values to the range $[-1, 1]$. Each Gaussian graph $\mathbf{A}^k = [\alpha_{ij}^k]$, corresponding to a unique perspective on class relationships, serves as the posterior graph in variational inference (we demonstrate in Section 3.4 that the choice of view does not affect the variational result). Finally, these graphs are used to produce a diverse set of fused features:

$$\mathbf{f}_i^k = (1 - \omega)(\mathbf{A}^k \mathcal{H}_b) + \omega \mathbf{h}_i, \quad \omega \in [0, 1), \tag{16}$$

where $\mathcal{H}_b$ denotes the base-class prototypes and $\omega$ controls the balance between integrated prior knowledge and the original features.

### 3.4 LEARNING OF HC-BDC

**Learning of Bayesian Relation Inference** In order to train the Bayesian relation inference component more effectively, inspired by VRNN Chung et al. (2015), we use a graph variational inference method to train the model. We use two random variables requiring optimization to describe the same stochastic process. Specifically, we treat any view of Gaussian graph as the posterior graph $q\left(\mathbf{M}^k, \mathbf{S}^k \mid \mathbf{h}_\text{i}, \mathcal{H}_\text{b}^e\right)$, where the random variables $\mathbf{M}^k$ and $\mathbf{S}^k$ can describe the same stochastic process that generate a Gaussian graph. The prior graph $p\left(\mathbf{M}^{(0)} \mid \mathbf{h}_\text{i}, \mathcal{H}_\text{b}^e\right)$ can also be represented by these two random variables through Gaussian graph transformation: $p\left(\mathbf{M}^{(0)}, \mathbf{S}^{(0)} \mid \mathbf{h}_\text{i}, \mathcal{H}_\text{b}^e\right)$. Our goal is to minimize the following KL divergence:

$$\text{KL}\left(q\left(\mathbf{M}^k, \mathbf{S}^k \mid \mathbf{h}_\text{i}, \mathcal{H}_\text{b}^e\right) \| p\left(\mathbf{M}^{(0)}, \mathbf{S}^{(0)} \mid \mathbf{h}_\text{i}, \mathcal{H}_\text{b}^e\right)\right). \tag{17}$$

In this equation, $m_{ij} \in \mathbf{M}$ is a sample from the binomial distribution $\mathcal{B}(n, \lambda)$ that can be approximated by a Gaussian proxy. Since each variable in $\mathbf{S}$ is affected by $\tilde{\alpha}$ in Eq.(12), we have $s_{ij} \mid \tilde{\alpha}_{ij} \sim \mathcal{N}\left(\tilde{\alpha}_{ij} * \mu_{ij}, \tilde{\alpha}_{ij} * \sigma_{ij}^2\right)$, the KL term can be further written as:

$$\mathcal{L}_{KL} = \underbrace{\text{KL}(\mathcal{B}(n, \lambda_{ij}) \| \mathcal{B}(n, \lambda_{ij}^{(0)}))}_{\text{Binomial term}} +$$

$$\mathbb{E}_{\widetilde{\alpha}_{ij}}[\text{KL}(\mathcal{N}(\widetilde{\alpha}_{ij} * \mu_{ij}, \widetilde{\alpha}_{ij} * \sigma_{ij}^2) \tag{18}$$

$$\underbrace{\| \mathcal{N}(\widetilde{\alpha}_{ij} * \mu_{ij}^{(0)}, \widetilde{\alpha}_{ij} * \sigma_{ij}^{(0)2}))]}_{\text{Gaussian term}}.$$

The calculation of this KL term does not depend on the random variable $\mathbf{S}$. By this equation we avoid the need to find the KL term for a large number of multi-view Gaussian graphs, which greatly simplifies the calculation. In Eq.(18), the Gaussian term can be calculated easily. According to recent research Liu & Jia (2023); Huang et al. (2020), we can use the Gaussian approximation of the binomial distribution to perform the approximate calculation. (The upper bound on the error of this approximation is guaranteed. For the detailed proof, please refer to the Appendix.).

**Learning of HC-BDC** In our framework, classification of the query fusion features is accomplished by training a simple logistic regression classifier. This classifier operates independently within each episode, undergoing complete parameter initialization and training adopting the fusion features of support set $\mathcal{F}_{\text{supp}}$. The optimization of HC-BDC combines the cross-entropy loss on the query set with the KL divergence loss $\mathcal{L}_{\text{KL}}$ from the Bayesian relation inference module:

$$\mathcal{L}\text{total} = \mathcal{L}_{\text{CE}} + \gamma \mathcal{L}_{\text{KL}}, \quad \gamma \in (0, 1]. \tag{19}$$

## 4 EXPERIMENTS

### 4.1 COMPARISON EXPERIMENTS

To validate the effectiveness of the proposed model, we compare it with different baseline methods for few-shot learning on *mini*Imagenet Russakovsky et al. (2015), tieredImagenet Ren et al. (2018b) and Dermnet 2022 dataset as shown in Table 1 and 2. The *mini*Imagenet and tieredImagenet datasets are standard few-shot datasets, while the Dermnet dataset is a long-tail dataset of dermatological images. For fair comparison, we use the same setting to conduct comparisons with state-of-the-art methods (SOTAs) on the three datasets. We adopt a ResNet-18 pre-trained model as backbone network in each dataset. The dataset is divided in the same way as our method is set up.

Table 1: Comparison of HC-BDC and baselines in accuracy on the *mini*Imagenet and tieredImagenet dataset under 5-way 1-shot/5-shot scenarios

| Method | *mini*Imagenet | | tieredImagenet | |
|---|---|---|---|---|
| | 1-shot(%) | 5-shot(%) | 1-shot(%) | 5-shot(%) |
| MAML 2017 | 48.70 | 63.11 | 60.85 | 78.82 |
| PN 2017 | 49.42 | 68.20 | 61.33 | 80.02 |
| MN 2016 | 43.44 | 55.31 | 62.80 | 81.16 |
| DC 2021 | 68.12 | 83.08 | 78.19 | 89.90 |
| Sum-min 2022 | 68.32 | 82.71 | 73.63 | 87.59 |
| DDWM 2023 | 68.58 | 84.65 | 80.02 | 87.85 |
| Sem-Few 2024 | 78.94 | 86.49 | 82.37 | 89.89 |
| AMU-Tuning 2024 | 80.57 | 88.55 | 83.75 | 93.10 |
| **HC-BDC (ours)** | **86.12**$_{\pm 0.18}$ | **97.17**$_{\pm 0.08}$ | **86.68**$_{\pm 0.22}$ | **95.31**$_{\pm 0.12}$ |

The value following the $\pm$ symbol represents the 95% confidence interval.

Table 2: Comparison of HC-BDC and baselines in accuracy on the Dermnet dataset under 5-way 1-shot/5-shot scenarios

| Method | 1-shot(%) | 5-shot(%) |
|---|---|---|
| MAML 2017 | 44.05 | 60.17 |
| PN 2017 | 43.76 | 60.22 |
| MN 2016 | 44.23 | 61.13 |
| DC 2021 | 48.99 | 66.75 |
| tSF 2022 | 49.38 | 68.15 |
| AMU-Tuning 2024 | 46.58 | 65.13 |
| GAP 2023 | 48.92 | 68.89 |
| **HC-BDC (ours)** | **52.20**$_{\pm 0.28}$ | **70.93**$_{\pm 0.13}$ |

The value following the $\pm$ symbol represents the 95% confidence interval.

The results demonstrate our model's state-of-the-art performance in both 5-way 1-shot and 5-way 5-shot scenarios. Traditional methods like MAML Finn et al. (2017), PN Snell et al. (2017) and MN Vinyals et al. (2016) achieve limited accuracy due to their inability to fully utilize available training data. While Distribution Calibration (DC) Yang et al. (2021) shows improved performance by generating relation graphs through Euclidean distance, its fixed metric fails to capture

complex inter-class relationships, leaving room for improvement in both accuracy and interpretability. Recent advances in few-shot learning Afrasiyabi et al. (2022); Kang et al. (2023); Lai et al. (2022); Wei et al. (2023); Zhang et al. (2024); Tang et al. (2024) have achieved notable performance improvements. However, these methods remain limited by (1) inadequate exploitation of base-class knowledge and (2) absence of human-like multi-perspective relational reasoning. Our HC-BDC framework addresses these limitations by integrating a gated mixture-of-experts mechanism for dynamic base-class knowledge selection, simulating fast-thinking processes, along with Bayesian multi-view Gaussian graphs that emulate slow-reasoning exploration. This cognitively-inspired dual-phase design supports comprehensive relationship modeling and efficient knowledge transfer, achieving state-of-the-art performance.

## 4.2 ABLATION STUDIES

### 4.2.1 ANALYSIS OF COMPONENT EFFECTIVENESS

To investigate the contribution of each component and the effectiveness of simulating human cognition, we design the ablation studies by constructing three variants of HC-BDC. Specifically, the three ablated models are constructed as follows: *w/o Bayes, MoE* removes both the Bayesian relation inference module and the MoE component, using only a single expert with a linear layer for feature fusion; *w/o Bayes* eliminates the Bayesian module while retaining the MoE-based fast thinking; and *w/o MoE* removes the mixture-of-experts component but preserves the slow-reasoning Bayesian relation inference. The performance of each variant on *mini*ImageNet is shown in Table 3.

Table 3: Performance comparison of HC-BDC and its variants on the *mini*Imagenet dataset.

| Variant of HC-BDC | 5-way 1-shot (%) | 5-way 5-shot (%) |
|:---:|:---:|:---:|
| *w/o Bayes, MoE* | 38.24 | 47.73 |
| *w/o Bayes* | 48.76 | 62.86 |
| *w/o MoE* | 84.76 | 96.80 |
| HC-BDC (ours) | **86.12** | **97.17** |

The results in Table 3 reveal that both the MoE fast-thinking component and the Bayesian slow-reasoning module contribute to model performance. Notably, removing the Bayesian component (*w/o Bayes*) leads to a more significant performance drop compared to removing the MoE component (*w/o MoE*). This aligns with the cognitive finding that the slow-reasoning system plays a critical role in reducing judgment errors Kahneman (2011). The MoE component also provides a clear benefit, supporting flexible knowledge association as in fast thinking. The full HC-BDC framework, integrating both cognitive mechanisms, achieves the best performance, validating the importance of simulating dual-phase human reasoning for few-shot learning.

### 4.2.2 PERFORMANCE WITH VARYING NUMBERS OF GAUSSIAN RELATION GRAPH VIEWS

The Bayesian relation inference module's core functionality is to infer class relationships from multiple perspectives. To validate the effectiveness of multi-view relational inference, we conduct experiments with varying numbers of Gaussian relation graph views ($N_{\text{Gaus}}$) ranging from 1 to 1000, as shown in Table 4.

The results demonstrate a clear performance improvement as the number of views increases, particularly when transitioning from few to moderate numbers of views. This confirms that multi-view relational inference is crucial for the model's effectiveness in few-shot classification tasks. At around 100 views, the model achieves an optimal balance between computational efficiency and classification accuracy, indicating it has sufficiently learned multi-perspective relationships at this point. Notably, we observe a slight performance degradation when increasing views from 500 to 1000 (87.21% to 87.06% for 5-way 1-shot), suggesting that 500 views may already provide sufficient diversity for learning inter-class relationships, while additional views might introduce redundant perspectives that slightly harm performance.

Table 4: Performance with varying numbers of Gaussian relation graph views

| Views | 5-way 1-shot (%) | 5-way 5-shot (%) |
|---|---|---|
| 1 | 33.09 | 84.62 |
| 2 | 52.72 | 92.24 |
| 5 | 70.54 | 96.67 |
| 10 | 77.81 | 96.40 |
| 20 | 83.67 | 97.16 |
| 50 | 85.49 | 96.54 |
| 100 | 86.12 | 97.17 |
| 200 | 86.61 | 96.63 |
| 500 | **87.21** | **97.18** |
| 1000 | 87.06 | 96.98 |

## 4.3 VISUAL ANALYSIS

Interpretability is crucial for relation inference methods, especially in sensitive domains such as medical diagnosis. To validate the reliability of our HC-BDC framework, we visualize relation graphs for images from both *mini*Imagenet and Dermnet datasets (Figure 2 and 3). For each image, we average the multi-view Gaussian graphs from the expert with highest attention weights. Relationship intensities between target and base classes are shown as heatmaps, with three strongly positive-correlated and three strongly negative-correlated base classes selected for comparison.

### 4.3.1 VISUALIZATION ON *mini*IMAGENET DATASET

We conduct visual analysis on two *mini*Imagenet image sets, each containing two images from the same class but with visual differences. The relation graphs show that HC-BDC yields interpretable reasoning results consistent with human cognition.

In Image Set 1, both target images depict the same bird species, and the model consistently identifies another bird category (Base 8) as strongly positively correlated. Positively correlated classes are mostly animals, while negative ones are typically man-made objects, aligning with human categorical reasoning. Moreover, relation graphs for different images of the same category show strong consistency: Base 8 and Base 10 maintain high positive correlations, while Base 4 and Base 7 show strong negative correlations. This intra-class consistency reveals the interpretability of our model's relation graphs.

Image Set 2 further demonstrates human-aligned reasoning: categories with strong positive correlations share morphological and color traits with the targets. Relation graphs for different target images within the same category also exhibit high consistency. These results confirm HC-BDC's ability to capture semantic relationships resembling human cognitive processes, offering essential explainability for few-shot classification decisions.

### 4.3.2 VISUAL ANALYSIS ON DERMNET DATASET

The target image in relation visualization 1, Figure 3 is *Perioral Steroid*, belonging to the *Acne and Rosacea* major class in the Dermnet dataset. The image shows a diffuse inflammatory response. The three most relevant base classes (*Atopic Dermatitis*, *Poison Ivy and other Contact Dermatitis*, and *Vasculitis*) each represent a distinct inflammatory type, lending high medical confidence to the model's relation graph. Notably, although the target's onset site is the lip, the model does not treat location as decisive for positive correlation. Meanwhile, the three strongest negatively correlated diseases exhibit clearly different symptoms and onset sites, indicating that the model does not rely primarily on lesion location.

In relation visualization 2, the target *tufted-folliculitis* belongs to *Alopecia and other Hair Diseases*. The strongest positively correlated base class also falls under this category, and both exhibit hair loss, showing a strong association. Conversely, the most negatively correlated base classes are visually distinct from the target.

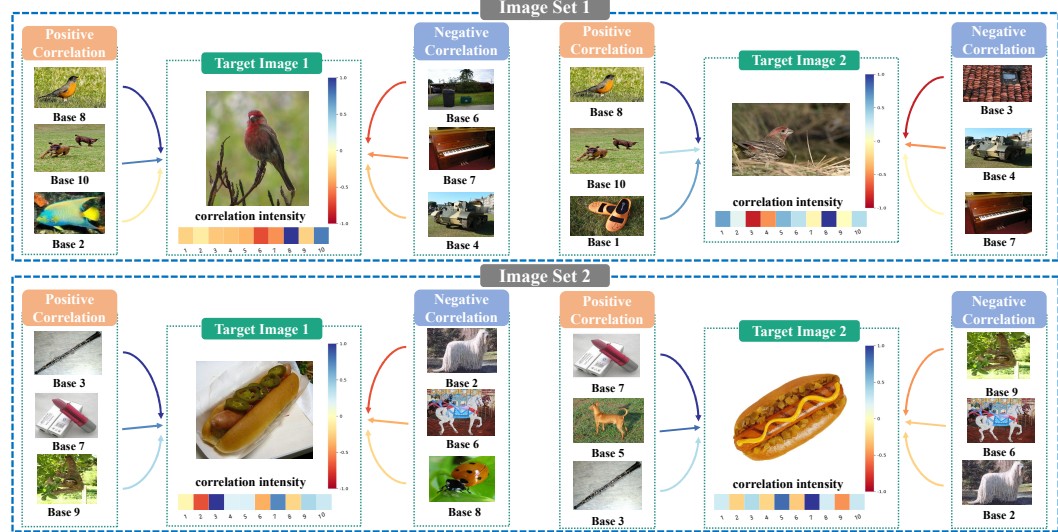

Figure 2: Visual analysis on *mini*Imagenet dataset. The heat map represents the associations between the target image and base classes, with brighter colours representing positive correlations and darker colours representing negative correlations.

These results confirm the model's interpretability and its ability to capture clinically meaningful inter-class relationships, supporting its applicability in medical scenarios.

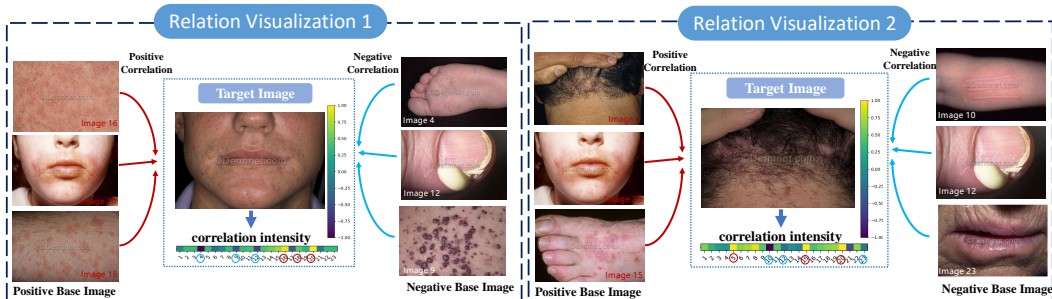

Figure 3: Visual analysis of HC-BDC on Dermnet dataset.

## 5 CONCLUSION

We propose **HC-BDC**, a human cognition-inspired few-shot learning framework that bridges machine learning and cognitive science by simulating dual-phase human reasoning. It integrates a fast-thinking MoE component for dynamic knowledge selection with a slow-reasoning Bayesian relation inference module for multi-view relational modeling, effectively capturing both adaptive knowledge retrieval and diverse associative capabilities characteristic of human cognition.

Extensive experiments on standard and medical benchmarks show that HC-BDC achieves state-of-the-art classification accuracy while providing interpretable, human-aligned relational graphs. Its balance of performance and explainability, underpinned by cognitive principles, offers strong potential for real-world applications and points toward a new direction for building human-like learning systems.

## 6 REPRODUCIBILITY STATEMENT AND ETHICS STATEMENT

We provide an open-source implementation of HC-BDC in the supplementary material, along with the dataset splitting code for Dermnet.

The authors do not foresee any negative social impacts of this work. All authors disclosed no relevant relationships.

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

## A    EXPERIMENT CONFIGURATION AND DETAILS

### A.1    PRELIMINARIES OF FEW-SHOT LEARNING

In the standard $n$-way $k$-shot classification setting, a model is presented with a support set $\mathcal{S} = (\mathbf{x}_i, \mathbf{y}_i)$ containing $k$ labeled examples for each of $n$ novel classes, and a query set $\mathcal{Q} = (\mathbf{x}_j)$ of unlabeled instances. The goal is to predict labels for query samples by leveraging limited information from the support set.

Our model employs $n$-way $k$-shot episodes for training, validation, and testing, while randomly selecting a set of classes from the training dataset as base classes.

### A.2    BASELINE METHODS DETAILS

To validate the effectiveness of the proposed model, we compare it with different baseline methods for few-shot learning. The baseline methods we compare are shown as follows:

- MAML Finn et al. (2017): An algorithm for meta-learning which is model-agnostic.

- Prototypical Networks (PN) Snell et al. (2017): A classical Metric-Based meta-learning method.

- Matching Networks (MN) Vinyals et al. (2016): A classical Metric-Based meta-learning method which uses LSTM to augment the network.

- Distribution Calibration (DC) Yang et al. (2021): A method for distribution calibration based on manually set Euclidean distances.

- PatchProto + tSF (tSF) Lai et al. (2022): A transformer-based semantic filter with Patch-Proto network for few-shot classification.

- GAP Kang et al. (2023): A meta-learning method with a geometry-adaptive preconditioner.

- Sum-min Afrasiyabi et al. (2022): A method for extracting and matching sets of feature vectors for few-shot image classification.

- DDWM Wei et al. (2023): An orientation-driven weighting method to make the feature distribution of a few-shot class accurately fit the true distribution.

- Sem-Few Zhang et al. (2024): A semantic-aided few-shot learning framework that employs a Semantic Evolution process to automatically generate high-quality semantics from class names.

- AMU-Tuning Tang et al. (2024): A CLIP-based few-shot learning method that learns effective logit bias by exploiting auxiliary features.

### A.3    TRAINING DETAILS

The details of the resources for training and the versions of the software are provided in Table 5.

Table 5: The hardware and software configuration for training.

|          |            |                                          |
|----------|------------|------------------------------------------|
|          | Python     | 3.10                                     |
| Software | PyTorch    | 1.13.1+cu116                             |
|          | numpy      | 1.24.3                                   |
|          | torchvision| 0.14.1+cu116                             |
|          | CPU        | Intel(R) Xeon(R) CPU E5-2680 v4 @ 2.40GHz|
| Hardware | RAM        | 128 GB                                   |
|          | GPU        | GeForce RTX 3090                         |

### A.4 Implementation details

For the training stage, we use the Adam optimizer and set the learning rate to 1**e**-4. We train the network for 1000 epochs and save the best performing model across 200 validation episodes for testing. The average accuracy of 1000 episodes is reported as the final result. The details of the hyperparameters are provided in Table 6.

Table 6: The configuration of hyper-parameters for training.

| Hyper-parameter | Value |
| --- | --- |
| N_Gaus | 100 |
| Edge_Dim | 128 |
| G_Dim | 256 |
| Batch_Size | 64 |
| Epoch | 1000 |
| Learning_Rate | 1**e**-4 |
| Lambda1 | 5**e**-4 |
| Lambda2 | 4**e**-4 |
| Weight_Decay | 3**e**-4 |
| N_Base | 10 |
| Experts | 1 |
| Omega | 0.5 |
| ValRuns | 200 |
| ImageSize | 224 |

### A.5 Code and Dataset

The HC-BDC model and the code for processing the Dermnet dataset are packaged in the supplementary materials.

We use *mini*Imagenet[1]Russakovsky et al. (2015), tieredImagenetRen et al. (2018b), and Dermnet[2] dataset for experiments. *mini*Imagenet dataset contains 100 classes with 600 images per class, and is divided into 64 base classes, 16 validation classes, and 20 novel classes in all experiments. tiered-Imagenet dataset contains 34 major categories, each containing 10 to 30 subcategories (i.e. classes, 608 subcategories in total). These are divided into 20 training categories, 6 validation categories, and 8 test categories. Dermnet dataset contains 23 broad classes of dermatology images and can be manually divided into more specific classes. The images in the dataset are in JPEG format, consisting of 3 channels, i.e., RGB. We divide the images in Dermnet dataset into secondary classes based on their names and removed all classes with less than 10 images to ensure that the 5way-5shot task could be completed. Finally, we divide the dataset into 344 classes with 17,206 images.The classes are randomly divided into training set, validation set and test set in the ratio of 7:1.5:1.5.

## B  Proofs of Key Theorems

### B.1  Theorem 1

Let $\mathcal{N}\left(\mu, \sigma^2\right)$ denotes a Gaussian distribution with $\mu < 1/2$, and let $\mathcal{B}(n, \lambda)$ denotes a Binomial distribution with $n \to +\infty$ and $\lambda \to 0$, where $n$ is increasing while $\lambda$ is decreasing. There exists a real constant $m$ such that if $m = n\lambda$ and if we define:

---

[1]https://paperswithcode.com/dataset/mini-imagenet
[2]https://www.kaggle.com/datasets/shubhamgoel27/dermnet

$$f_1(x) = \text{KL}\left(\mathcal{N}(x, x(1-x)) \| \mathcal{N}\left(\mu, \sigma^2\right)\right)$$
$$f_2(x) = \text{KL}(\mathcal{N}(x, x(1-x)) \| \mathcal{N}(n\lambda, n\lambda(1-\lambda)))$$
$$f_2^* = \min_x f_2(x), \text{ where } x \in (0, 1)$$

according to exist works Huang et al. (2020), we have that: $f_1(x)$ attains its minimum on the interval $(0, 1)$ and $f_2(x) - f_2^*$ is bounded on the interval $(0, \sqrt{2}/2 - 1/2)$, with:

$$x = m = \tfrac{1+l-\sqrt{1+l^2}}{2}, \text{ where } l = \tfrac{2\sigma^2}{1-2\mu}$$

Suppose we are given a Gaussian distribution $\mathcal{N}\left(\tilde{\mu}_i, \tilde{\sigma}_i^2\right)$, whose parameter $\tilde{\mu}_i$ is specifically parameterized by the neural network that can guarantee $\tilde{\mu}_i < 1/2$. By De Moivre-Laplace theorem, we have that $\mathcal{N}\left(n\lambda_i, n\lambda_i\left(1 - \lambda_i\right)\right)$ is a good approximation for $\mathcal{B}\left(n, \lambda_i\right)$. They are asymptotically equivalent as $n$ increases. Let $m_i = n\lambda_i$, direct parameterization of both the infinite parameter $n$ and the near-zero parameter $\lambda_{i,j}$ can be avoided by adopting a re-parametrization trick Kingma & Welling (2013). This trick draws samples from such Binomial distribution via its Gaussian proxy $\mathcal{N}\left(m_i, m_i\left(1 - m_i\right)\right)$.

## B.2 THEOREM 2

Suppose we are given two Binomial distributions, $\mathcal{B}(n, \lambda)$ and $\mathcal{B}\left(n, \lambda^0\right)$ with $n \to +\infty, \lambda^0 \to 0$ and $\lambda \to 0$, where $n$ is increasing while $\lambda$ and $\lambda^0$ are decreasing. There exists a real constant $m$ and another real constant $m^{(0)}$, such that if $m = n\lambda \, and \, m^{(0)} = n\lambda^{(0)}$ and if $\lambda > \lambda^0$, we have:

$$\text{KL}\left(\mathcal{B}(n, \lambda) \| \mathcal{B}\left(n, \lambda^0\right)\right) < m \log \frac{m}{m^{(0)}}$$
$$+ (1 - m) \log \frac{1 - m + m^2/2}{1 - m^{(0)} + m^{(0)^2}/2}$$

By Theorem 2 which is proofed in previous work Huang et al. (2020), we have a closed-form solution that is irrelevant to $n$ for the ELBO.

## B.3 PROOF OF THEOREM 1

Let $\mathcal{N}\left(\mu, \sigma^2\right)$ denotes a Gaussian distribution with $\mu < 1/2$, and let $\mathcal{B}(n, \lambda)$ denotes a Binomial distribution with $n \to +\infty$ and $\lambda \to 0$, where $n$ is increasing while $\lambda$ is decreasing. There exists a real constant $m$ such that if $m = n\lambda$ and if we define:

$$f_1(x) = \text{KL}\left(\mathcal{N}(x, x(1-x)) \| \mathcal{N}\left(\mu, \sigma^2\right)\right)$$
$$f_2(x) = \text{KL}(\mathcal{N}(x, x(1-x)) \| \mathcal{N}(n\lambda, n\lambda(1-\lambda))$$
$$f_2^* = \min_x f_2(x), \text{ where } x \in (0, 1)$$

we have that: $f_1(x)$ attains its minimum on the interval $(0, 1)$ and $f_2(x) - f_2^*$ is bounded on the interval $(0, \sqrt{2}/2 - 1/2)$, with:

$$x = m = \tfrac{1+l-\sqrt{1+l^2}}{2}, \text{ where } l = \tfrac{2\sigma^2}{1-2\mu}$$

Proof. The derivative of the function $f_1(x)$ over $x$ can be written as:

$$f_1'(x) = x^2 - \left(1 + \frac{2\sigma^2}{1 - 2\mu}\right) x + \frac{\sigma^2}{1 - 2\mu}$$

We set it as 0 and solve for $x$, giving:

$$x = \begin{cases} \frac{1+l-\sqrt{1+l^2}}{2} & \text{if } \mu < 1/2 \\ \frac{1+l+\sqrt{1+l^2}}{2} & \text{if } \mu > 1/2 \end{cases}, \text{ where } l = \frac{2\sigma^2}{1-2\mu} \qquad (1)$$

Let $x = n\lambda$, the function $f_2(x)$ can be written as:

$$f_2(n\lambda) = \sqrt{\frac{1-n\lambda}{1-\lambda}} + \frac{1-\lambda}{2(1-n\lambda)} - 1/2$$

Let $g(n\lambda) = \lim_{\lambda \to 0} f_2(n\lambda)$, we have:

$$g(n\lambda) = \sqrt{1-n\lambda} + \frac{1}{2(1-n\lambda)} - 1/2$$

Let $z = \sqrt{1-n\lambda}$, we have:

$$g(z) = z + 1/\left(2z^2\right) - 1/2$$

The derivative of function g(z) over $z$ can be written as:

$$g'(z) = 1 - 1/z^3$$

Given that $z \in (0,1)$, we have $g'(z) < 0$. Then $g(z)$ attains its minimum 1 when $z$ approaches 1. Equivalently, $f_2(n\lambda)$ attains its minimum 1 when $n\lambda$ approaches 0.

Considering Eq.(1), we find that $n\lambda$ is bounded on $(0, 1/2)$ if $\mu < 1/2$, We then calculate the difference between $f_2(n\lambda)$ and its minimum. It can be written as:

$$\begin{aligned} \Delta f_2(n\lambda) &= \lim_{\lambda \to 0} \left[ f_2(x) - f_2^* \right] \\ &= g(n\lambda) - 1 \\ &= \sqrt{1-n\lambda} + \frac{1}{2(1-n\lambda)} - 3/2 \end{aligned}$$

Let $m = n\lambda$, the derivative of function $\Delta f_2(m)$ over $m$ can be written as:

$$\Delta f_2'(m) = \frac{1 - (1-m)^{3/2}}{2(1-m)^2} > 0$$

Then $\Delta f_2(m)$ is monotonically increasing over $(0, 1/2)$. Therefore $\Delta f_2(m)$ is bounded on $(0, \sqrt{2}/2 - 1/2)$.

## C   SUPPLEMENTARY EXPERIMENT RESULTS AND FURTUER DISCUSSION

### C.1   ABLATION STUDIES

#### C.1.1   EXPERT CONFIGURATION

In the MoE Relation Inference Component, we configure $N_E$ experts, each containing $N_B$ base classes. This component simulates the selective attention mechanism in human fast thinking, where each expert maintains a set of prior knowledge (base classes) that may have varying degrees of association with novel input categories. To investigate how the granularity of prior knowledge partitioning and the coverage of prior knowledge affect model performance, we conduct two ablation

Table 7: Performance of HC-BDC under different expert-base configurations (Total base classes fixed at 30)

| Experts $\times$ Bases | 5-way 1-shot (%) | 5-way 5-shot (%) |
|---|---|---|
| $1 \times 30$ | 80.53 | 95.48 |
| $3 \times 10$ | 86.12 | 97.17 |
| $5 \times 6$ | 86.75 | 97.01 |
| $6 \times 5$ | 86.88 | 97.37 |
| $10 \times 3$ | 86.96 | 97.48 |
| $30 \times 1$ | 87.52 | 97.49 |

studies focusing on: (1) the number of experts and (2) the base class allocation per expert. The results are presented in Tables 7 and 8.

Table 7 shows the results when fixing the total amount of prior knowledge (30 base classes) while distributing them among varying numbers of experts. The results demonstrate that overly coarse allocation (using only one expert) leads to significant performance degradation. While gradually increasing expert specialization improves model accuracy marginally, it comes with substantially increased computational overhead due to each expert simulating a slow-thinking process. To balance computational cost and model accuracy, we ultimately adopt a configuration with 3 experts, each containing 10 base classes. This setup better aligns with human fast-thinking characteristics - being unconscious (each expert maintains broad generalization) and rapid (fewer experts).

Table 8: Performance of HC-BDC with varying numbers of experts (10 base classes per expert)

| Experts | 5-way 1-shot (%) | 5-way 5-shot (%) |
|---|---|---|
| 1 | 84.76 | 96.80 |
| 2 | 84.96 | 96.78 |
| 3 | 86.12 | 97.17 |
| 4 | 85.47 | 96.64 |
| 5 | 86.33 | 97.18 |

Table 8 presents the results when fixing the number of base classes per expert while varying the total number of experts. Increasing experts from 1 to 3 improves 5-way 1-shot accuracy by 1.36% (from 84.76% to 86.12%), while further expansion to 5 experts yields only 0.21% additional gain. This suggests that increasing base class coverage effectively improves model accuracy when prior knowledge is scarce. However, beyond a certain threshold, additional base classes provide diminishing returns, possibly because excessive base classes make it harder for the model to identify the most relevant knowledge.

### C.1.2 ABLATION STUDY ON CLASSIFIER SELECTION

The final few-shot classification is accomplished by training a lightweight classifier on the enriched fusion features. We evaluate several classifier options to identify the most suitable one for our model, including Support Vector Machine (SVM), Random Forest (RF), K-Nearest Neighbors (KNN), Neural Network (NN), and Logistic Regression (LR), with results presented in Table 9.

The results reveal significant performance variations across different classifiers. Both the neural network and logistic regression achieve superior performance, with logistic regression slightly outperforming the neural network while being more computationally efficient. Other traditional methods (SVM, RF, KNN) show notably lower accuracy. Based on these findings, we select logistic regression as our final classifier due to its optimal balance between performance and efficiency.

Table 9: Performance comparison of different classifiers

| Classifier | 5-way 1-shot (%) | 5-way 5-shot (%) |
|---|---|---|
| SVM | 80.52 | 96.41 |
| RF | 75.31 | 95.02 |
| KNN | 70.44 | 85.51 |
| NN | 86.02 | 96.92 |
| LR | 86.12 | 97.17 |

### C.1.3 MORE INTUITIVE VISUAL ANALYSIS

In the proposed model, we use classes with a large amount of data as the base classes, which have the advantage that the features of the base classes can portray the overall distribution of the classes well. However, since there are also large differences between the images within each category (e.g., differences in the site of onset or even symptoms), this approach to base class selection results in the generated relation between target and base classes being less intuitively, and some relation ambiguity may occur. To solve this problem, we consider setting the base classes to some specific images to obtain more intuitive inter-class relations. Specifically, we selected the most representative image from each of the base classes with more obvious differences from other classes as the base classes, and infer the relation through these images. For any input image of the target class, the generated relation intensity graph is the relation between that image and the base class images, which can be more intuitively expressed through visual analysis of the relations found by the Bayesian relation inference module. The visualization result is shown in Figure 4 and the accuracy result on 5-way 1-shot and 5-way 5-shot tasks is shown in Table 10.

In the upper part of Figure 4, we show the results of the visual analysis using single image as a base classes. We average the multi-view Gaussian graphs and visualize the relationship between the target image and different base class images in the form of heatmaps, and select three images with strong positive and negative correlations respectively for further visualization analysis. It is worth noting that the target picture and the picture with serial number 16 belong to the same category in the Dermnet dataset, which indicates that the proposed HC-BDC can effectively capture the potential relations between different objects. As can be seen from Table 10, using a single image as the base classes on the 5-way 1-shot and 5-way 5-shot tasks has a small decrease in accuracy (about 3% at 5-way 1/5-shot) compared to using a large number of images as base classes, which indicates that the proposed method does not require a large number of images for base classes. A small number of base class images can also provide a good distributional calibration for the target imags.

### C.1.4 VISUAL ANALYSIS ON ROBUSTNESS

In the proposed model, we use base classes that are strongly related to the task (e.g., for the Dermnet skin disease dataset, we use some of the dermatology classes in the dataset as base classes). In practice, there may not be enough data to be used as base classes, so the performance of the model in the absence of data strongly related to the task as base classes is important. To explore the robustness of the Bayesian Distribution Calibration model, we replace the base classes with animal data that are not relevant to skin diseases. This replacement aims to explore the ability of the Bayesian relation inference component to infer potential relations between target class and base classes that is very different from the target class. The visualization result is shown in Figure 4 and the accuracy result on 5-way 1-shot and 5-way 5-shot tasks is shown in Table 10.

The bottom half of Figure 4 shows the visualization result using animal image data as base classes. From the result, it is seen that the categories that have strong correlations with the target images are cat, horse and tiger. Intuitively, these three base categories have strong visual similarities, where the two categories of cat and tiger belong to the same family of felines, which can prove that the proposed Bayesian distribution calibration model is able to capture potential relations of the different categories. As can be seen from Table 10, the results using animal data as base classes still achieve high level of accuracy, which can prove that the proposed model is robust to the selection of base classes.

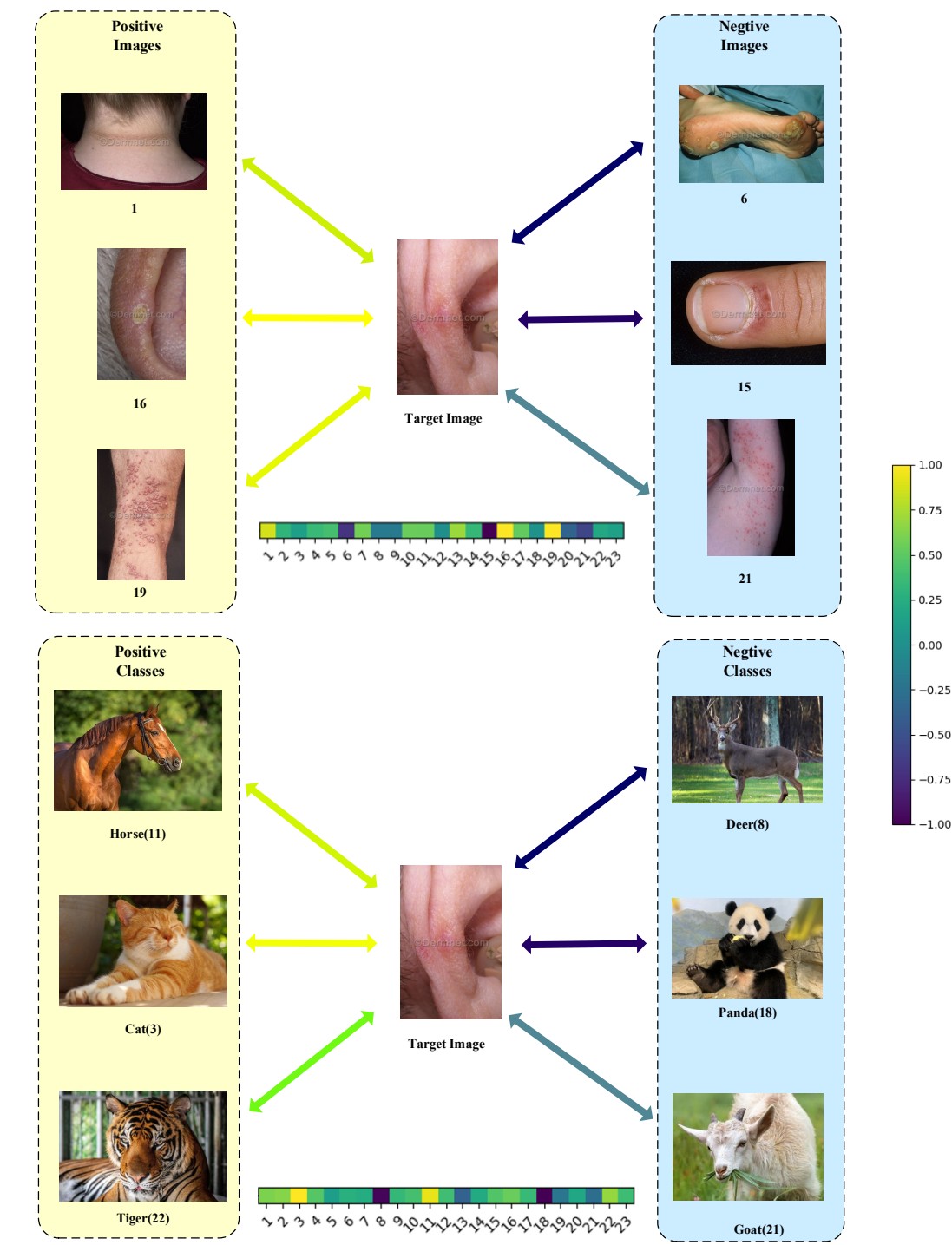

Figure 4: Schematic representation of the results of relation inference obtained using single image or animal data.

.

### C.1.5 COMPARISON OF CONVENTIONAL ALGORITHMS AND FEW-SHOT LEARNING ALGORITHMS

In few-shot learning algorithms, training a model using conventional algorithms can be difficult due to the large number of categories and the fact that the data for most of the categories is scarce. We

Table 10: Performance of HC-BDC on Dermnet dataset with various base classes

| Method | 5way1shot(%) | 5way5shot(%) |
|---|---|---|
| HC-BDC + Single image | 49.56 | 68.04 |
| HC-BDC + Animal image | 48.99 | 67.25 |
| **HC-BDC(Ours)** | **52.20** | **70.93** |

divide the test set of the Dermnet dataset in a ratio of 8:2 into a new training and test set. Then we generate a conventional algorithm model by replacing the few-shot classification part of the Bayesian relation inference model (Multi-view Gaussian graph generation component and logistic regression classifier) with a linear classification head. We freeze the Bayesian relation inference module and fine-tune the classification head on the new training set and test it on the new test. We compare the accuracy of this algorithm with that of the few-shot learning algorithms on the 5-way 1-shot task. The experiment result is shown in Table 11.

Table 11: Comparison of few-shot algorithms and conventional algorithm on Dermnet dataset

| Method | Acc(%) |
|---|---|
| MAML | 44.05 |
| PN | 43.76 |
| MN | 44.23 |
| DC | 48.99 |
| tSF | 49.38 |
| GAP | 48.92 |
| *HC-BDC + Conventional Algorthms* | *43.50* |
| **HC-BDC(Ours)** | **52.20** |

Specifically, we adopt a three-layer artificial neural network as the linear classification head of the conventional algorithm, trained for 1500 epochs using the Adam optimizer with the learning rate of 0.0008. The result shows that the conventional algorithm's accuracy is similar to that of the early few-shot algorithms on the 5-way 1-shot task. It is worth noting that traing the conventional algorithm is time-consuming and overall performs less well than the few-shot algorithms.

### C.1.6 COMPARISON EXPERIMENTS ON CUB DATASET

We further compare the proposed model with different baseline methods for few-shot learning on CUB Wah et al. (2011) dataset. The results are shown in Table 12.

Table 12: Comparison of HC-BDC and baselines in accuracy on the CUB dataset

| Method | 5way1shot(%) | 5way5shot(%) |
|---|---|---|
| MAML 2017 | $50.45_{\pm0.97}$ | $59.60_{\pm0.84}$ |
| PN 2017 | 72.99 | 86.64 |
| MN 2016 | 73.49 | 84.45 |
| DC 2021 | 78.29 | 88.92 |
| Sum-min 2022 | 79.60 | 90.48 |
| DDWM 2023 | 80.40 | 90.75 |
| **HC-BDC(Ours)** | **80.72** | **91.40** |

Due to limited arithmetic, we reduce the number of episodes tested on this dataset to 60 and report the average accuracy as the result of our model. From the results, our model performs slightly better than the baseline method we compared it to. In addition to the fact that we did not carefully set the hyperparameters of the model, the reason for the lower performance of HC-BDC on the CUB

dataset may be that the CUB dataset is a specialized bird classification dataset. Since the differences between various birds are much smaller compared to the different categories in *mini*Imagenet, it may make it more difficult for the model to correctly infer relations between these classes.

# D  POSSIBLE QUESTIONS ABOUT THIS PAPER

**Q1: What's downstream task and "conventional algorithms"?** The downstream task represents the few-shot classification in this paper and our model can be used for other few-shot downstream tasks. Conventional algorithm denotes traditional deep learning method rather than few-shot learning method.

**Q2: Is there more testimony on the viewpoint that manually set relations are often incomplete and biased?** Previous studies illustrate that manually set relations are biased. Adam Santoro et al. noted that practitioners define the relations between symbols using the language of logic and mathematics. But *manual approaches are not robust to relational questions.* Besides, the underlying structure is characterized by *sparse but complex relations, which is very difficult to model accurately by manual methods* Santoro et al. (2017).

We also illustrate experimentally that manually set relations do not perform as well as the proposed automatic relation inference method. The DC method in the comparison experiments is similar to the proposed method in terms of model architecture, and the *DC method uses manually set Euclidean distances in comparing relations between classes and our HC-BDC method outperforms it.*

**Q3: Are there specific indicators to assess the interpretability of the relation intensity graphs generated?** There are no specific indicators or expert-assessed inter-class relations to quantitatively analyze our relation graphs. However, *the corresponding broad class has strong positive correlation with most target images (e.g., acne-cystic in Acne).* For Figure 4 in the appendix, *we can also intuitively find some possible relations*: tigers and cats both positively correlated with the target image and belong to the same family of felids, which are biologically strongly positively correlated. These can indicate that our HC-BDC can capture the potential relations between classes.

**Q4: How are images converted into graphs?** We refer to existing works on graph generation methodsLiu & Jia (2023); Huang et al. (2020). Specifically, node embeddings are the *feature of a single target image or average features of each base classes generated by the backbone network.* The *Bayesian relation inference component automatically generates relations between nodes.*

**Q5: The limitations of the work.** There are no specific indicators or expert-assessed inter-class relations to quantitatively analyze our relation graphs. Besides, since we are not professional dermatologists, the visual analysis of the Dermnet dataset may not be interpreted from a very specialized point of view, but only from a generalist's thinking, which may lead to an incomplete description of the interpretability of the model. In addition, the generation of summary graphs is based on the ideas presented in existing studies that humans engage in a myriad of unconscious perceptions when performing relation thinking which can be viewed as a sampling of a binomial distribution with $n \to \infty$ and $\lambda \to 0$ Huang et al. (2020) instead of generating them without prior knowledge.

