# OpenReview forum: "HC-BDC: Human Cognition-Inspired Bayesian Distribution Calibration for Few-Shot Classification"
_ICLR.cc/2026/Conference — ICLR 2026 Conference Withdrawn Submission_

### Official Review · Reviewer_V1no · 2025-10-16

**Soundness:** 3
**Presentation:** 2
**Contribution:** 3
**Rating:** 6
**Confidence:** 4

**Summary:**

The paper presents a novel framework for few-shot image classification that integrates a fast thinking and slow reasoning mechanism. The approach combines Mixture-of-Experts (MoE) based knowledge routing with Bayesian inference, offering a promising balance between efficiency and interpretability. The proposed method is conceptually appealing and theoretically grounded, and the experimental results demonstrate strong performance across several benchmarks.

**Strengths:**

1. Novel conceptual framework: The integration of fast thinking (MoE routing) and slow reasoning (Bayesian inference) is innovative and provides an intuitive analogy to human cognition.

2. Theoretical soundness: The method is supported by a clear theoretical foundation and formal analysis.

3. Comprehensive experiments: Extensive evaluations across multiple datasets verify the effectiveness and robustness of the proposed approach.

**Weaknesses:**

1. Citation format: The paper’s citation style and formatting need correction to meet standard academic requirements.

2. Organization: The paper structure could be improved, particularly the Related Work section, which currently lacks logical flow and clear categorization.

3. Incomplete training details: The training procedure of the MoE component is not fully described, making it difficult to reproduce or assess the scalability and stability of the method.

**Questions:**

Given the extreme gap between base and novel classes, the MoE may lack sufficient knowledge of novel classes during testing. How does the proposed method address this domain generalization issue or ensure effective expert routing for novel class adaptation?

---

### Official Review · Reviewer_trwT · 2025-10-22

**Soundness:** 1
**Presentation:** 3
**Contribution:** 2
**Rating:** 2
**Confidence:** 4

**Summary:**

This paper introduces HC-BDC, a few-shot learning framework inspired by human cognitive processes, specifically the dual systems of "fast thinking" and "slow reasoning". The method employs a MoE model to simulate fast, intuitive knowledge routing and a Bayesian relation inference module to emulate slower, deliberate reasoning for calibrating feature distributions. The authors claim that this approach achieves new SOTA performance on several FSL benchmarks, including a particularly striking result of 97.17% on the miniImageNet 5-way 5-shot task, which I see as a result of data leakage.

**Strengths:**

1. The paper's motivation, which attempts to connect the architecture to human cognitive processes like "fast and slow thinking", is a creative and interesting way to frame the problem, even if the connection feels a bit forced.

2. The paper validates the model's reasoning via visual analysis, showing that the learned relations can be qualitatively aligned with human-understandable visual or semantic concepts.

**Weaknesses:**

1. As can be seen by the submitted code, the authors used a standard ImageNet-1k pre-trained backbone for evaluation on miniImageNet/tieredImageNet, which constitutes a critical data leakage. The test classes are not "novel" to the feature extractor. This invalidates the main performance claims and makes any comparison to legitimate FSL methods meaningless.
2. The authors claim "fair comparison," but this is questionable on multiple fronts. They use a 224x224 image resolution rather than the traditional 84x84, it is unclear if all compared methods use the same. More importantly, the comparison is fundamentally unfair because their model benefits from data leakage.
3. The RELATED WORKS section primarily cites papers that are 4-5 years old. Also, the paper fails to situate itself against or compare with more recent state-of-the-art methods, including those from 2025.
4. The proposed model is quite complex, appearing to be a simple mixture of several existing ideas. The "slow reasoning" component, in particular, seems heavily based on the work of Huang et al. (2020), which diminishes the novelty of the technical contribution.
5. The brain-inspired story, while a strength in its novelty, feels somewhat forced rather than being a guiding principle from the ground up.
6. Table 6 in the appendix lists the number of "Experts" as 1. This appears to be a typo.

**Questions:**

1. The ablation study in Table 3 shows that removing the MoE component (w/o MoE) results in only a minor performance drop, suggesting its contribution is marginal. This seems to undermine the "dual-system" hypothesis. I recommend that the authors clarify the parameter overhead introduced by the MoE module. Is the negligible performance gain worth its inclusion and the added complexity, or would a simpler model without the MoE be preferable?
2. The core issue with this paper is the data leakage from the pre-trained backbone. I suggest the author to re-run the key experiments on miniImageNet and tieredImageNet using a backbone that is trained exclusively on the designated base classes of those datasets. This would provide a fair evaluation of your method's true few-shot learning capability and is necessary to establish any credible performance claims.

---

### Official Review · Reviewer_ae68 · 2025-11-01

**Soundness:** 2
**Presentation:** 3
**Contribution:** 2
**Rating:** 2
**Confidence:** 4

**Summary:**

The paper proposes HC-BDC, a cognitively-inspired method for few-shot image classification. The method introduces a two-stage pipeline:

(1) a fast-thinking Mixture-of-Experts (MoE) module that routes query features toward selected base-class prototypes, and

(2) a slow, Bayesian reasoning module that builds multi-view Gaussian relational graphs to calibrate the class distribution and generate fusion features for classification.

Experiments are reported on miniImageNet, tieredImageNet, and a custom Dermnet medical dataset, showing large performance gains over prototypical/meta-learning baselines. Several ablations demonstrate contributions of MoE routing, multi-view Gaussian calibration, and Bayesian inference. The paper also includes interpretability visualizations and some theoretical motivation for the Gaussian approximation.

**Strengths:**

1. Clear conceptual framing connecting "fast/slow" human cognition with a two-component model (MoE + Bayesian inference).

2. Strong reported empirical numbers on 5-way miniImageNet/tieredImageNet benchmarks, greatly outperforming standard few-shot baselines.

3. Ablation studies explore number of Gaussian views, expert count, classifier choice, and base knowledge selection.

**Weaknesses:**

**Outdated and insufficient evaluation protocol**

The paper evaluates exclusively on 5-way k-shot episodic benchmarks (miniImageNet, tieredImageNet)—a setting widely used before 2020, but now considered outdated in the era of large-scale pretrained models (CLIP, ALIGN, Meta-Transformer, etc.). Since the proposed method already relies on a pretrained backbone, the correct comparison is few-shot transfer on full ImageNet-1K (1/2/4/8/16-shot), not toy 5-way meta-learning splits. Modern few-shot methods (CoOp, CoCoOp, CLIP-Adapter, Tip-Adapter, KgCoOp, VPT, MaPLe, FD-Align) use this protocol and dramatically change the baseline landscape. No comparison is made to these methods, making it impossible to judge whether HC-BDC is competitive in 2024–2025.

**Baselines are outdated**

The core comparison set (ProtoNet, MatchingNet, MAML, etc.) does not match the current state of the field. No ViT-based, CLIP-based, adapter-based, or prompt-based baselines are included. As a result, the paper compares to 2017–2019 meta-learning models rather than 2022–2025 few-shot models.

**Questions:**

1. Why is the evaluation limited to 5-way k-shot instead of the standard 1/2/4/8/16-shot full ImageNet-1K protocol used by CoOp, Tip-Adapter, KgCoOp, etc.?

2. Does HC-BDC scale to 1,000-class few-shot transfer, or is it only feasible in 5-way episodic classification?

---

### Note · Authors · 2025-11-28

I have read and agree with the venue's withdrawal policy on behalf of myself and my co-authors.